# MECHANISTIC PERMUTABILITY: MATCH FEATURES ACROSS LAYERS

**Nikita Balagansky[1,2]\*, Ian Maksimov[3,1], Daniil Gavrilov[1]**
[1] T-Tech, [2] Moscow Institute of Physics and Technologies, [3] HSE University

## ABSTRACT

Understanding how features evolve across layers in deep neural networks is a fundamental challenge in mechanistic interpretability, particularly due to polysemanticity and feature superposition. While Sparse Autoencoders (SAEs) have been used to extract interpretable features from individual layers, aligning these features across layers has remained an open problem. In this paper, we introduce SAE Match, a novel, data-free method for aligning SAE features across different layers of a neural network. Our approach involves matching features by minimizing the mean squared error between the folded parameters of SAEs, a technique that incorporates activation thresholds into the encoder and decoder weights to account for differences in feature scales. Through extensive experiments on the Gemma 2 language model, we demonstrate that our method effectively captures feature evolution across layers, improving feature matching quality. We also show that features persist over several layers and that our approach can approximate hidden states across layers. Our work advances the understanding of feature dynamics in neural networks and provides a new tool for mechanistic interpretability studies.

## 1 INTRODUCTION

In recent years, foundation models have become pivotal in natural language processing research (Llama Team, 2024; Team et al., 2024). As these models are applied to a growing range of tasks, the need to interpret their predictions in human-understandable terms has intensified. However, this interpretability is challenged by the presence of polysemantic features—features that correspond to multiple, often unrelated concepts—especially prevalent in large language models (LLMs).

A significant advancement in addressing polysemanticity is the *superposition hypothesis*, which posits that models represent more features than their hidden layers can uniquely encode. This leads to features being entangled in a non-orthogonal basis within the hidden space (Bricken et al., 2023; Arora et al., 2018; Elhage et al., 2022), complicating interpretation because individual neurons or features do not correspond neatly to singular, human-understandable concepts.

To unravel this complexity, *Sparse Autoencoders* (SAEs) trained on the hidden states of LLMs were employed, using feature sizes significantly larger than the models' hidden dimensions (Yun et al., 2021; Bricken et al., 2023). SAEs aim to extract monosemantic features—sparse activations that occur only when processing specific, interpretable functions. For instance, Templeton et al. (2024) used this approach to interpret the Claude Sonnet 3 model, while Lieberum et al. (2024) open-sourced an SAE for the Gemma 2 model, enabling researchers to delve deeper into LLM interpretability.

While SAEs have advanced the interpretability of individual model layers, a crucial question remains: How do these interpretable features evolve throughout the layers of a model during evaluation? Understanding this evolution is essential for a comprehensive interpretation of the model's internal dynamics and decision-making processes.

In this paper, we introduce **SAE Match**, a data-free method for aligning SAE features across different layers of a neural network. Our approach enables the analysis of feature evolution throughout the model, providing deeper insights into the internal representations and transformations that occur as data propagate through the network. By addressing the challenge of feature alignment across layers,

---

\*n.n.balaganskiy@tbank.ru

we contribute to a more comprehensive understanding of neural network behavior and advance the field of mechanistic interpretability.

Our main contributions are as follows:

- We propose **SAE Match**, a novel method for aligning Sparse Autoencoder features across layers without the need for input data, enabling the study of feature dynamics throughout the network.
- We introduce *parameter folding*, a technique that incorporates activation thresholds into the encoder and decoder weights, improving feature matching by accounting for differences in feature scales.
- We validate our method through extensive experiments on the Gemma 2 language model, demonstrating improved feature matching quality and providing insights into feature persistence and transformation across layers.

By advancing methods for feature alignment and interpretability, our work contributes to the broader goal of making neural networks more transparent and understandable, facilitating their responsible and effective deployment in various applications.

## 2 BACKGROUND

The goal of the mechanistic interpretability field is to develop tools for understanding the behavior of neural networks. The main challenge with naïve model interpretation approaches stems from the polysemanticity of features in trained models, which states that each feature is linked to a combination of unrelated inputs (Bricken et al., 2023). One explanation for polysemanticity is superposition hypothesis, which suggests that the number of features trained by a model exceeds the number of neurons in that model (i.e., the size of its hidden layer) (Arora et al., 2018; Elhage et al., 2022). In this scenario, the features might be represented using a non-orthogonal basis in the hidden space, making them challenging to interpret with simple methods.

Assuming the superposition hypothesis is correct, Bricken et al. (2023) used Sparse Autoencoders (SAEs) (Yun et al., 2021) to uncover features in language models that are comprehensible to humans. Given the assumption that models capture more features than their hidden dimensionality allows, unveiling these features can be achieved by training an autoencoder on the hidden states of a model with a large representation size (Templeton et al. (2024) used representation sizes orders of magnitude larger than a model's hidden size to interpret the Claude 3 Sonnet model):

$$
\begin{aligned}
\boldsymbol{f}(\boldsymbol{x}) &= \sigma\left(\boldsymbol{W}_{\text{enc}}\boldsymbol{x} + \boldsymbol{b}_{\text{enc}}\right), \\
\hat{\boldsymbol{x}}(\boldsymbol{f}) &= \boldsymbol{W}_{\text{dec}}\boldsymbol{f} + \boldsymbol{b}_{\text{dec}}, \\
\boldsymbol{f}(\boldsymbol{x}) &\in \mathbb{R}^F, \boldsymbol{x} \in \mathbb{R}^d, F \gg d.
\end{aligned}
\tag{1}
$$

Here, $\sigma$ is an activation function (e.g., ReLU). Importantly, $\boldsymbol{f}(\boldsymbol{x})$ is made to be sparse to help the SAE capture monosemantic features. The final loss is defined as $\mathcal{L}(\boldsymbol{x}) = \text{MSE}(\boldsymbol{x}, \hat{\boldsymbol{x}}(\boldsymbol{f}(\boldsymbol{x}))) + \lambda \mathcal{L}_{reg}$. The sparsity of the model can be increased by adjusting the $\lambda$ hyperparameter, which balances the trade-off between the reconstruction term and the regularization term. However, determining the optimal value for $\lambda$ is not straightforward.

Building on this approach, Lieberum et al. (2024) introduced a family of SAE models trained on the hidden states of the Gemma 2 model for each layer. These models, along with detailed feature descriptions, are available on Neuronpedia[1]. Notably, this work departs from previous methods by employing JumpReLU (Rajamanoharan et al., 2024) as the activation function:

$$
\text{JumpReLU}(\boldsymbol{z}) = \boldsymbol{z} \odot H(\boldsymbol{z} - \boldsymbol{\theta}), \boldsymbol{\theta} \in \mathbb{R}^F;
\tag{2}
$$

where $H(\cdot)$ is the Heaviside step function, and $\boldsymbol{\theta}$ represents learnable thresholds for each component of $\boldsymbol{z}$.

---

[1] https://www.neuronpedia.org/gemma-2-2b

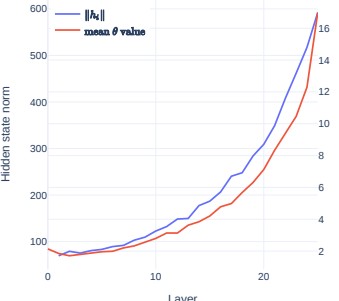 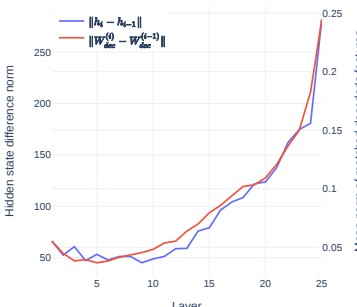

Figure 1: **Left:** Hidden state norms and the mean $\boldsymbol{\theta}$ value in trained JumpReLU activations within SAE modules. **Right:** Dynamics of hidden state norm changes and the differences in norms of matched decoder columns $\boldsymbol{W}_{\mathrm{dec}}$ after the folding operation. These results suggest that $\boldsymbol{\theta}$ captures the growth of hidden state norms. After folding $\boldsymbol{\theta}$ into the weights, the decoder weights $\boldsymbol{W}'_{\mathrm{dec}}$ become dependent on the dynamics of hidden state norms, leading to a lower overall MSE during matching. For more details on the folding operation and method description, see Section 3.

## 3 MATCHING SPARSE AUTOENCODER FEATURES

While SAEs can extract human-interpretable features from specific layers of a language model, understanding how these features evolve across layers remains an open question. To address this, we introduce **SAE Match**, a data-free method to align SAE features across different layers. This approach enables us to analyze the evolution of features throughout the model's depth, providing deeper insights into the model's internal representations.

Our central idea is that features from different layers might be similar but permuted differently—that is, the $i$-th feature in layer $A$ might correspond to the $j$-th feature in layer $B$. Therefore, aligning features across layers involves finding the correct permutation that matches semantically similar features.

**Hypothesis 1:** *Features $\boldsymbol{f}^{(A)}$ from layer $A$ can be matched with features $\boldsymbol{f}^{(B)}$ from layer $B$ by calculating the mean squared error (MSE) between the relevant SAE weights. These weights might be either the rows of the encoder weights $\boldsymbol{W}_{\mathrm{enc}}^{(A)}$ and $\boldsymbol{W}_{\mathrm{enc}}^{(B)}$ or the columns of the decoder weights $\boldsymbol{W}_{\mathrm{dec}}^{(A)}$ and $\boldsymbol{W}_{\mathrm{dec}}^{(B)}$ (or both). Rows or columns that have a low MSE between them suggest semantic similarity between the features.*

This matching problem resembles the task of aligning weights from two neural networks that perform the same function but have permuted weights (Ainsworth et al., 2022). In our case, we aim to find a permutation matrix $\boldsymbol{P}^{(A \to B)} \in \mathcal{P}_F$ (the set of permutation matrices) that aligns the features of layer $A$ with those of layer $B$. Formally, we seek a permutation that minimizes the MSE between the decoder weights:

$$\boldsymbol{P}^{(A \to B)} = \arg\min_{\boldsymbol{P} \in \mathcal{P}_F} \sum_{i=1}^{d} \|\boldsymbol{W}_{\mathrm{dec}_{i,:}}^{(A)} - \boldsymbol{P}\boldsymbol{W}_{\mathrm{dec}_{i,:}}^{(B)}\|^2 = \arg\max_{\boldsymbol{P} \in \mathcal{P}_F} \left\langle \boldsymbol{P}, \left(\boldsymbol{W}_{\mathrm{dec}}^{(A)}\right)^\top \boldsymbol{W}_{\mathrm{dec}}^{(B)} \right\rangle_F, \quad (3)$$

where $\langle \boldsymbol{A}, \boldsymbol{B} \rangle_F = \sum_{i,j} \boldsymbol{A}_{i,j} \boldsymbol{B}_{i,j}$ is the Frobenius inner product. To solve this, one can utilize a Linear Assignment Problem (LAP) solver (Ainsworth et al., 2022). The same logic applies to the encoder weights, provided they are transposed.

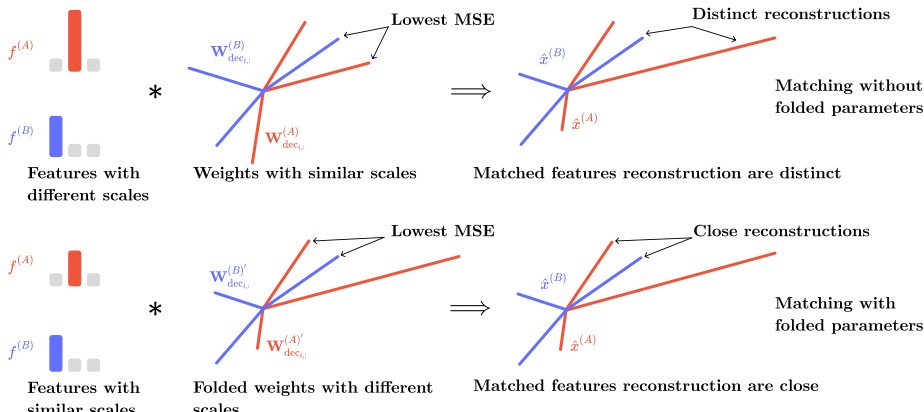

Figure 2: A schematic illustration of the differences in SAE matching with and without folded parameters. When no folding is performed (**top**), $\boldsymbol{\theta}$ encapsulates differences in hidden state norms, causing features $\boldsymbol{f}^{(A)}$ and $\boldsymbol{f}^{(B)}$ to have different scales, while the columns of decoder weights $\boldsymbol{W}_{\text{dec}_{i,:}}^{(A)}$ and $\boldsymbol{W}_{\text{dec}_{i,:}}^{(B)}$ have similar norms. Matching similar columns leads to differences in the actual reconstructions of the input $\hat{\boldsymbol{x}}$, which we hypothesize is detrimental for matching SAE features. With $\boldsymbol{\theta}$ folding (**bottom**), we transfer the differences in input (and thus feature) norms to the decoder weights $\boldsymbol{W}_{\text{dec}_{i,:}}^{(A)'}$ and $\boldsymbol{W}_{\text{dec}_{i,:}}^{(B)'}$, thereby matching features while accounting for differences in input norms. As a result, reconstructions of matched features are closer to each other than in the unfolded variant of the algorithm. See Section 3.1 for more details.

## 3.1 FOLDING SPARSE AUTOENCODER WEIGHTS

While the above method can be directly applied, it does not utilize the information stored in the learnable thresholds $\boldsymbol{\theta}$ of the JumpReLU activation function. Moreover, differences in feature scales can lead to suboptimal matching when minimizing MSE between vectors with varying norms. To address this, we propose a technique called **parameter folding**, where we incorporate the activation thresholds into the encoder and decoder weights:

$$\boldsymbol{W}_{\text{enc}}' = \boldsymbol{W}_{\text{enc}} \operatorname{diag}\left(\frac{1}{\boldsymbol{\theta}}\right), \ \boldsymbol{b}_{\text{enc}}' \ = \boldsymbol{b}_{\text{enc}} \odot \frac{1}{\boldsymbol{\theta}}, \ \boldsymbol{W}_{\text{dec}}' = \boldsymbol{W}_{\text{dec}} \operatorname{diag}(\boldsymbol{\theta}), \ \boldsymbol{\theta}' \ = \mathbf{1}, \qquad (4)$$

where $\operatorname{diag}(\boldsymbol{\theta})$ creates a diagonal matrix from $\boldsymbol{\theta}$, and $\odot$ denotes element-wise multiplication. This transformation does not alter the SAE's output but adjusts the weights to account for differences in feature scales.

Our hypothesis is that parameter folding helps align decoder weights in a way that reflects the actual dynamics of hidden state norms during model evaluation (see Figure 1). We observed that $\boldsymbol{\theta}$ values, in fact, encapsulate the growth of hidden state norms. By folding $\boldsymbol{\theta}$ into the weights, we normalize the decoder weights to align with these dynamics. This process allows us to match decoder weights based on their MSE in the space of actual hidden state norms. By folding parameters, we can match features across layers in a data-free manner, without needing input data to capture input scales. For a schematic illustration of behavior, refer to Figure 2

**Hypothesis 2:** *Folding the activation thresholds into the weights improves the quality of matching.*

By incorporating the folding process, we ensure that the comparison of decoder weights is sensitive to the scale of the hidden states, rather than relying solely on angular proximity. This allows for a more accurate reflection of the underlying feature dynamics and enhances the model's ability to effectively match features across layers. Consequently, by aligning decoder weights to hidden state norms, we achieve a more robust evaluation of feature similarity, improving the overall performance of the Sparse Autoencoder model.

## 3.2 COMPOSING PERMUTATIONS

When a permutation is obtained from layer A to layer B, and another from layer B to layer C, it allows us to approximate a permutation from layer A to layer C by composing these two permutations $\boldsymbol{P}^{(A \to C)} \approx \boldsymbol{P}^{(B \to C)} \boldsymbol{P}^{(A \to B)}$ instead of evaluating actual permutation between layers $A$ and $C$. Extending this to a model with $T$ layers, we define all possible permutation paths as: $\mathbb{P} = \{\boldsymbol{P}^{(i \to j)} | j \in [1; T], i \in [0; j)\}$.

**Hypothesis 3:** *We can approximate the matching between any two layers, A and C, by composing permutations. However, the quality of this approximation decreases as the relative distance between these layers increases.*

By examining these composed permutations, we aim to uncover deeper insights into feature dynamics—how certain features retain or transform their semantic meanings as they pass through multiple layers of the network.

## 4 EXPERIMENTAL SETUP

We conducted experiments using Sparse Autoencoders (SAEs) trained on the Gemma 2 model, with feature descriptions from Neuronpedia generated by GPT-4o. We focused on SAEs with descriptions provided by Neuronpedia (for a complete list of SAEs used, refer to Appendix Table 1). For matching we used MSE from both decoder and encoder layers. During our initial experiments, we observed that the decoder-only option performs similarly to our scheme, while the encoder-only suffers from poor quality of matching (see Appendix Figure 12 for comparison). A more detailed analysis of the differences in matching using various sets of weights is deferred to future work.

To measure the quality of feature matching, we evaluated the quality of the match using two key metrics:

1. Mean Squared Error (MSE): The error between the permuted parameters of the matched features.
2. An external large language model (LLM) compared Neuronpedia descriptions of matched features, categorizing them as "SAME" (identical meanings), "MAYBE" (possibly similar meanings), or "DIFFERENT" (distinct meanings). Each experiment involved approximately 1,600 LLM evaluations over 100 feature paths spanning 16 layers (details in Appendix Section C).

When assessing how approximating hidden states with matched features affects language modeling performance, we used:

- Change in Cross-Entropy Loss ($\Delta L$): The difference in loss (in nats) between the model using the encode-permute-decode operation and the original model.
- Explained Variance: Calculated as $\mathrm{Var}(\hat{\boldsymbol{x}}^{(t+1)} - \boldsymbol{x}^{(t+1)}) / \mathrm{Var}(\boldsymbol{x}^{(t+1)})$. This metric assesses how well the approximated hidden state $\hat{\boldsymbol{x}}^{(t+1)}$ estimates the true hidden state $\boldsymbol{x}^{(t+1)}$, providing insight into the accuracy of the approximation.
- Matching Score: The probability of paired feature activation between two matched layers.

We tested our methods on subsets of OpenWebText (Gokaslan et al., 2019), Code[2], and WikiText (Merity et al., 2016). From each dataset, we randomly sampled 100 examples, truncated them to 1,024 tokens, and excluded the beginning-of-sequence (BOS) token when calculating metrics.

## 5 EXPERIMENTS

### 5.1 UNDERSTANDING PARAMETER FOLDING

We start by exploring parameter folding. To understand it, we used our method to create permutation matrices $\boldsymbol{P}^{(t-1 \to t)}, t \in [1; 26]$ based on the MSE values for folded and unfolded parameters.

---

[2]https://huggingface.co/datasets/loubnabnl/github-small-near-dedup

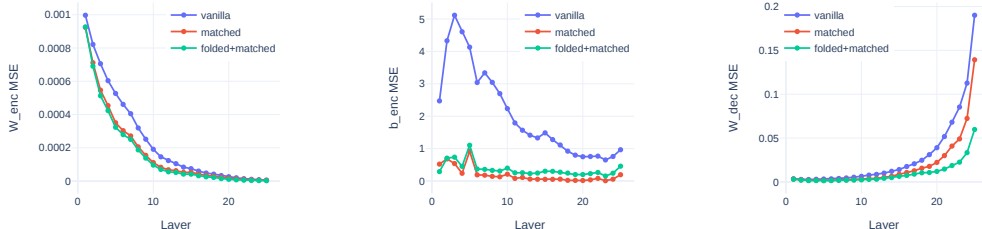

Figure 3: Results of the MSE objective for different layer matching methods. "Vanilla matching" refers to matching without any permutations. The "Matched" and "Folded+Matched" variants correspond to unfolded and folded matching, respectively. In all cases, MSE is evaluated with folded parameters (i.e., for unfolded matching, parameters are first matched, then folded, and finally MSE is evaluated). When considering input scales differences (see Section 3.1), this can be interpreted as the MSE in the scale of actual input reconstructions in the relevant layers. The unfolded matching consistently showed higher MSE in this scale, supporting Hypothesis 2. Note that $b_{\text{dec}}$ is omitted as it does not affect the order of features in the SAE layer. For further details, refer to Section 5.1.

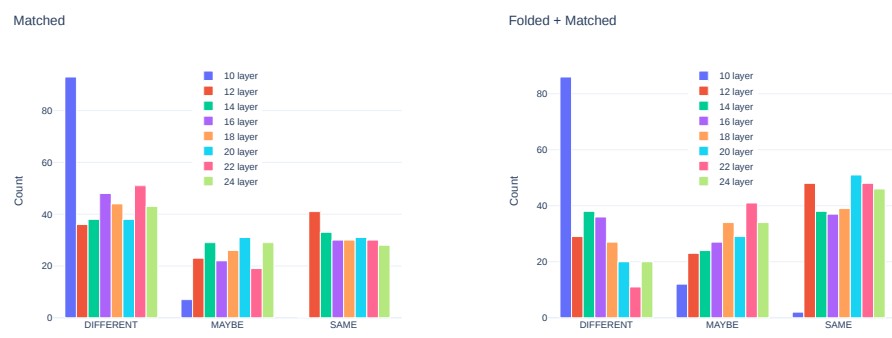

Figure 4: External LLM evaluation split by layers. As before - folding thresholds results in an optimistic labeling, it also affects deeper layers making them also more optimistic.

As Hypothesis 2 relies on the observation that $\boldsymbol{\theta}$ incapsulates differences in hidden state norms, we evaluated MSE of obtained permutations. More concretely, we evaluated MSE for folded matching, and for unfolded and naive (without any permutations) matching strategies, for which we first permuted SAE weights based on unfolded MSE values, and then evaluated MSE of appropriate folded weights. See Figure 3 for the results. Initially, naive matching displayed high MSE across all SAE parameters, indicating the lack of assurance in maintaining feature order between layers. Unfolded matching showed higher MSE in the scale of hidden state representations. Also note that since MSE is plotted with folded parameters, the MSE of the Encoder layer decreases, while the MSE of the Decoder layer increases (this behavior is explained by Equation 4).

We compared the performance of folded and unfolded matching based on semantic similarity with an external LLM. The results can be seen in Figure 4 for LLM evaluation across different layers. Folded matching exhibited improved feature matching quality compared to the unfolded variant. This result supports Hypothesis 2 regarding the behavior of matching with folded parameters.

Also, we observed a performance drop at the 10-th layer, which we study in Section 5.3.

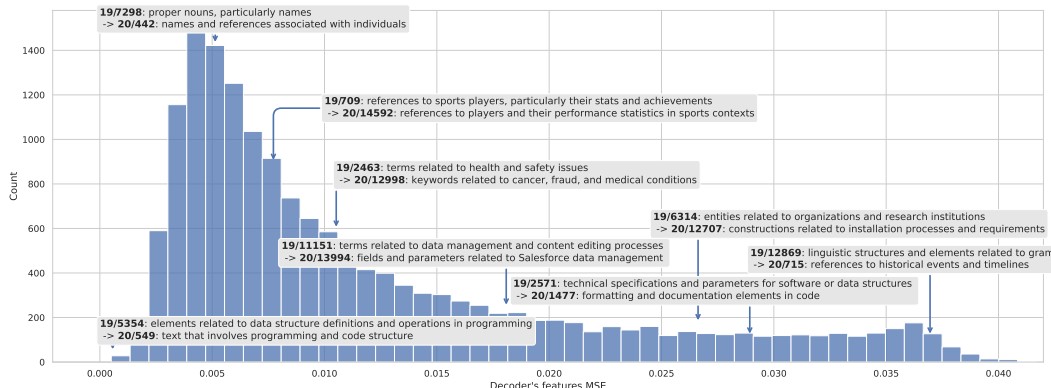

Figure 5: Features matched with folded parameters from the 19th to the 20th layer using the proposed method are sorted by their MSE values across the relevant SAE decoder weights. Features with small MSE values (on the left) indicate semantic similarity, while those with large MSE values (on the right) indicate that no similar features were found. For further details, refer to Section 5.2.

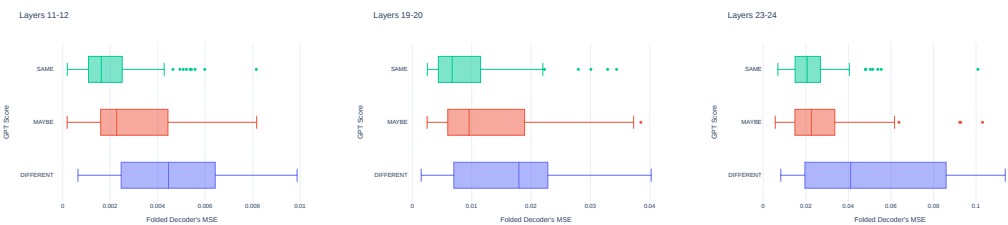

Figure 6: The distribution of MSE values with folded matching, grouped by three labels provided by an external LLM, is presented. Lower MSE values between pairs of layers indicate better matching. Note that the MSE value distribution varies across different layer pairs due to differences in weight scales For further analysis, refer to Section 5.2.

## 5.2 FEATURES MATCHING

Once we estabilished that folded parameters provide more accurate feature matching, we dive into exploring feature matching itself. We tested our hypothesis that features from different layers can be matched by calculating the mean squared error (MSE) between the parameters of Sparse Autoencoders (SAE). In this experiment, we focused on matching the 19-th and 20-th layers using parameter folding. The results are shown in Figure 5. Low MSE values indicated semantic similarity between features, while high MSE values suggested differences.

We also evaluated the distribution of MSE values for three pairs of layers matched with folded parameters, grouped according to external LLM evaluation. The results, shown in Figure 6, similarly revealed that lower MSE values were associated with the semantic similarity of matched features. It is important to note that the distribution of MSE values across different layer pairs varied due to differences in weight scales (as described in Section 3.1). See Appendix Section B for experiments with MSE thresholds.

Together, these results supported Hypothesis 1, confirming that MSE reflects feature similarity and enabling us to further investigate feature matching.

## 5.3 ON THE PERFORMANCE OF MATCHING ACROSS LAYERS

In previous sections, we observed a decline in the quality of feature matching evaluated by an external LLM in the lower layers of the model, particularly up to the 10th layer. To investigate this phe-

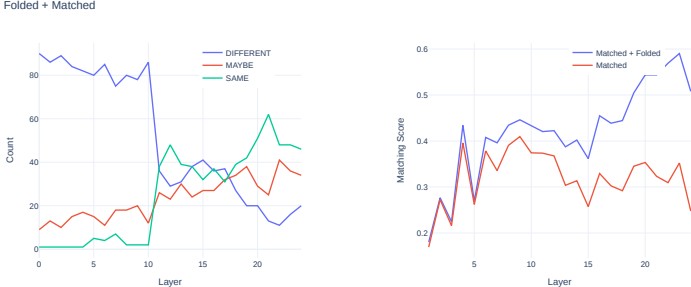

Figure 7: **Left:** External LLM evaluation of feature matching across layers with folded matching. **Right:** Matching Score across layers and across different matching methods. Folding parameters has a positive effect on Matching Score. The LLM evaluation shows near-zero "SAME" features in the initial layers, increasing after the 10th layer. The Matching Score shows a similar pattern but does not drop to zero in the initial layers, indicating some predictive ability. These results suggest that initial layers may have more polysemantic features, making them harder to evaluate using Neuronpedia descriptions. See Section 5.3 for more details.

nomenon, we extended our evaluation of folded matching across all layers of the model—excluding the final layer responsible for mapping to the vocabulary. We also examined the Matching Score metric (as defined in Section 4) to gain additional insights into the matching performance.

The results are presented in Figure 7. We observe that the number of "SAME" features—those with identical or closely related meanings—remains near zero in the initial layers and begins to increase after the 10th layer. A similar trend is seen in the Matching Score, although it does not drop to zero in the early layers, suggesting some ability to predict features from previous matched features.

Based on these results, SAE Match fails to align similar features in the initial layers. Our main explanation for this phenomenon lies in the differences in $l_0$ norms at these early layers (see Figure 15 for more details). We observe that when utilizing canonical SAEs, modules in the initial layers exhibit large variance in $l_0$ values. If we utilize SAEs with similar norms at each layer, the Explained Variance improves significantly. While we continue our experiments with canonical SAEs for consistency, this result suggests that these layers should be selected more precisely in future studies.

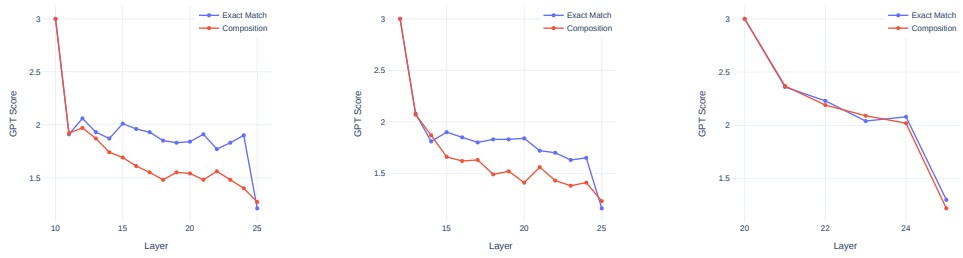

Figure 8: Results of the external LLM evaluation. We compare features from 10th, 12th and 20th layers with matched features in subsequent layers. It can be observed that feature similarity gradually declines, but remains significant for approximately five layers. We use 1 for DIFFERENT features, 2 for MAYBE features, and 3 for SAME features, and then average this score across layers. See Section 5.4 for more details.

### 5.4 Feature Persistence

In purpose to study dynamics of the feature from layer to layer we decided to conduct additional experiments with feature semantics persistence along the feature path.

We used three starting layers 10, 12, and 20 to evaluate permutations with all subsequent layers. We matched features between layers by both evaluating full permutation matrix (**Exact**) and by approximating it with composition (**Composition**) of composite permutation matrices (see Section 3.2). We then evaluated these permutations with an external LLM.

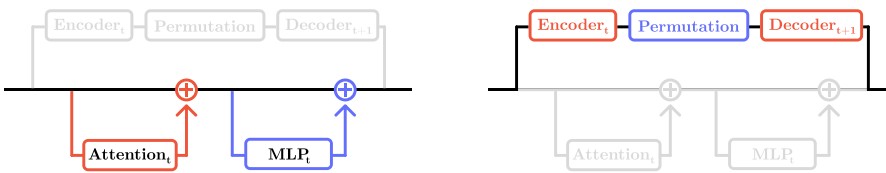

Figure 9: Schematic illustration of layer pruning with matched features. Instead of the standard evaluation (left), we compute features at layer $t$, apply the permutation to match them to layer $t + 1$, and then reconstruct the hidden state for layer $t + 1$, effectively skipping layer $t$. See Section 5.5 for more details.

See Figure 8 for the results. We observed that composition of permutations showed adequate results for nearby SAEs especially on the later layers, then it starts to diverge, which supports Hypothesis 3. Also, features were well matched when evaluating full permutation matrices, supposing that in later model layers features between layers are similar, which aligns with observations from Section 5.3.

### 5.5 SAE Match as a Layer Pruning

To further evaluate the effectiveness of our method, we conducted experiments where we pruned layers between matched Sparse Autoencoders (SAEs). The key assumption is that features remain consistent during a one-layer forward pass. If the permutation mapping between SAEs is accurate, we can approximate the output of a pruned layer using an encode-permute-decode operation:

$$
\begin{aligned}
\boldsymbol{f}^{(t)}(\boldsymbol{x}) &= \sigma\left(\boldsymbol{W}_{\text{enc}}^{(t)}\boldsymbol{x}^{(t)} + \boldsymbol{b}_{\text{enc}}^{(t)}\right), \\
\hat{\boldsymbol{f}}^{(t+1)} &= \boldsymbol{P}_t^{t+1}\boldsymbol{f}^{(t)}, \\
\hat{\boldsymbol{x}}^{(t+1)} &= \boldsymbol{W}_{\text{dec}}^{(t+1)}\hat{\boldsymbol{f}}^{(t+1)} + \boldsymbol{b}_{\text{dec}}^{(t+1)}.
\end{aligned}
\tag{5}
$$

Figure 9 provides a schematic illustration of this layer pruning method. See Figure 10 for the results.

We observed the smallest performance drop on the OpenWebText dataset, while a decrease in quality was noted at the last layer of the Code dataset. These findings indicate that the matched features share similar semantics, enabling us to approximate the next hidden states for all layers beyond the 10th layer. Also see Figure 11 for the Explained Variance and the change in loss across layers when using matching strategies, as well as with plain encoding and decoding of hidden states. We observed that at lower layers, matching yields lower explained variance compared to encoding and decoding the previous hidden state. This can be explained by errors introduced by the matching algorithm at these layers.

## 6 Limitations and Conclusion

We introduced **SAE Match**, a novel data-free method for aligning Sparse Autoencoder (SAE) features across neural network layers, tackling the challenge of understanding feature evolution amid polysemanticity and feature superposition. By minimizing the mean squared error between folded

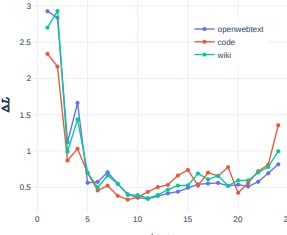 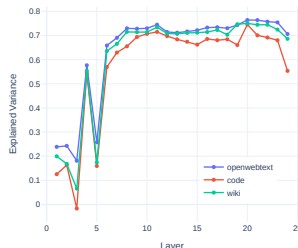

Figure 10: **Left:** Change in cross-entropy loss ($\Delta L$) after pruning each layer using matched features. **Right:** Explained variance of the approximated hidden states compared to the true hidden states. Starting from the 10th layer, pruning results in minimal performance loss, indicating that matched features effectively approximate the skipped layers. See Section 5.5 for more details.

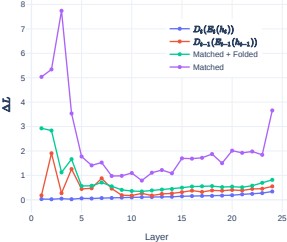 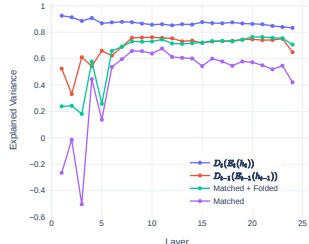

Figure 11: **Left:** The change in cross-entropy loss ($\Delta L$) with each layer of the SAE. **Right:** The explained variance of the approximated hidden states for each layer of the SAE. We present results for plain encoding and decoding of the current and previous hidden states, as well as for matching in both folded and unfolded variants. While folded matching eventually matches the results of encoding and decoding the previous hidden state, at lower layers it introduces errors that reduce performance.

SAE parameters—which integrate activation thresholds into the weights—we accounted for feature scale differences and improved matching accuracy. Experiments on the Gemma 2 language model validated our approach, showing effective feature matching across layers and that certain features persist over multiple layers. Additionally, we demonstrated that our method enables approximating hidden states across layers. This advances mechanistic interpretability by providing a new tool for analyzing feature dynamics and deepening our understanding of neural network behavior. We also showed that this approach can be applied not only to SAEs trained with the Gemma 2 model (see Figures 13, 19.

In the current version of the paper, we utilized only a scheme where we matched features based on all weights of the SAE (encoder, decoder, and biases). Future studies may include a more precise understanding of the behavior of different weights within the module. As we pointed out in Section 5.3, selecting appropriate SAEs based on their $l_0$ value is also a topic worth discussing. This value can be seen not only as a trade-off between sparsity and reconstruction quality, but also as a factor that affects matching features across layers. While in this work we focused on bijective matching to perform a step-by-step understanding of SAE features, it is evident that some features are not matched in this way. Therefore, one may want to explore more sophisticated methods to understand feature dynamics.

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

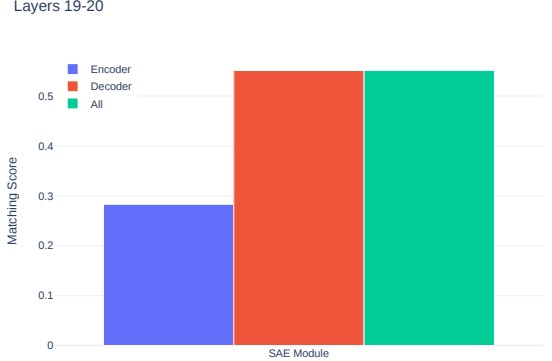

Figure 12: Matching Score evaluation of feature matching with different modes: encoder-only, decoder-only, and encoder-decoder matching; evaluated between 19-th and 20-th layers. Encoder-only suffers from poor performance, while decoder-only performs on par with the encoder-decoder scheme.

## A  ON THE MATCHING WITH DIFFERENT SPARSITY

In this section, we investigate how varying the sparsity level of Sparse Autoencoders (SAEs) affects the quality of feature matching across layers. The Gemma Scope release (Lieberum et al., 2024) provides SAEs trained with different levels of sparsity, measured by the mean $l_0$-norm of their

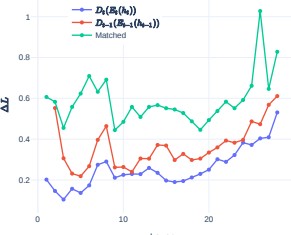 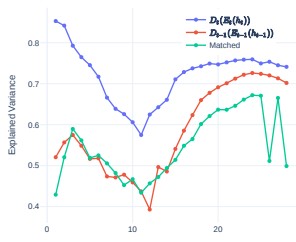

Figure 13: **Left:** The change in cross-entropy loss ($\Delta L$) with each layer of the SAE. **Right:** The explained variance of the approximated hidden states for each layer of the SAE. Results are for LLAMA-3.1-8B model

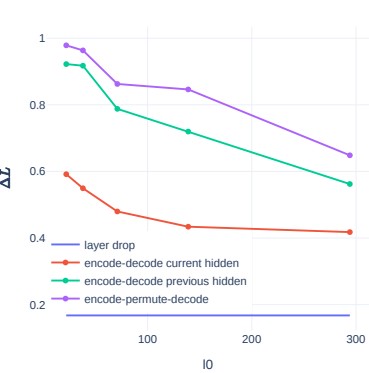 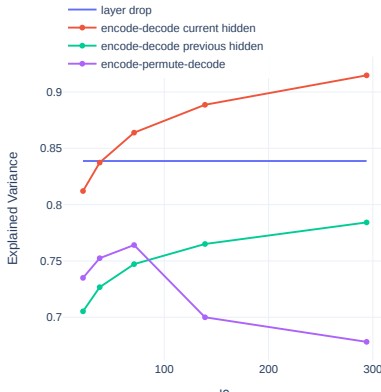

Figure 14: **Left:** Change in cross-entropy loss ($\Delta L$) with respect to the sparsity level (mean $l_0$-norm) of the SAE. **Right:** Explained variance of the approximated hidden states at different sparsity levels. Interestingly, explained variance has a peak on value at $l_0 \approx 70$. See Section A for discussion.

activations. Understanding the impact of sparsity is crucial because it influences the number of active features and, consequently, the effectiveness of our matching method.

To evaluate our approach under different sparsity conditions, we selected five SAEs trained on the 20th and 21st layers of the Gemma 2 model, each with a different mean $l_0$-norm. We applied our feature matching method to these SAEs and measured performance using the metrics described in Section 4. Specifically, we assessed how well the matched features could approximate the hidden states when a layer is omitted.

- **layer drop**: We skip the $t$-th layer entirely, using the hidden state $x^{(t)}$ as input for the $(t+1)$-th layer.
- **encode-decode current hidden**: use SAE reconstruction $\hat{x}^{(t)}$ as hidden state at $t$-th layer instead of $x^t$.
- **encode-decode previous hidden**: use SAE reconstruction from $t$-th layer $\hat{x}^{(t)}$ as a hidden state $x^{(t+1)}$ and omit evaluation of $t$-th layer.

First, when analyzing the change in cross-entropy loss ($\Delta L$), we observe that higher mean $l_0$-norm values correspond to a lower difference in loss across all methods. This is expected because SAEs with higher sparsity (i.e., lower $l_0$-norm) may not reconstruct the hidden states effectively, leading

to a higher $\Delta L$. Conversely, SAEs with lower sparsity (higher $l_0$-norm) capture more information, resulting in better approximations and lower $\Delta L$.

Notably, using features from lower layers to approximate subsequent layers leads to higher $\Delta L$ compared to both the encode-decode current and encode-decode previous hidden schemas. This indicates that directly reconstructing the hidden state of a layer using its own SAE provides better performance than attempting to predict it from earlier layers.

Regarding the explained variance—which measures how well the approximated hidden state $\hat{x}^{(t+1)}$ captures the true hidden state $x^{(t+1)}$—we observe that it peaks when the mean $l_0$-norm is approximately 70. This suggests that there is an optimal sparsity level where feature matching is most effective. At lower $l_0$-norms (higher sparsity), SAEs may fail to reconstruct hidden states adequately due to insufficient active features. On the other hand, when the number of active features exceeds 100, accurately matching all features between layers becomes more challenging. This can lead to the inclusion of noisy features, which negatively impacts the estimation of the hidden state, pushing it in the wrong direction. As other schemas do not utilize matching, we do not observe peak as for matching plot.

This experiment underscores a key insight of our work: the sparsity level of SAEs plays a crucial role in the success of data-free feature matching across layers. It emphasizes that not only does our method effectively align features across layers, but it also allows for the identification of optimal model configurations that balance sparsity and performance, further advancing the interpretability and efficiency of language models.

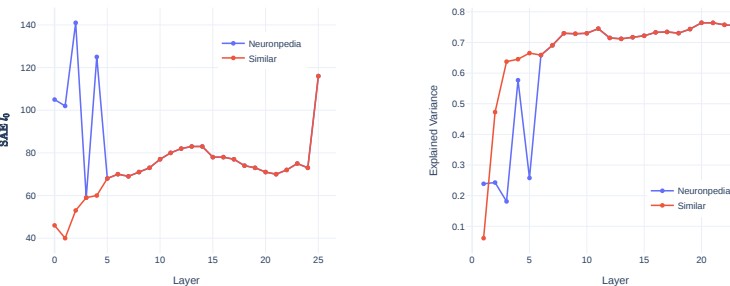

Figure 15: **Left:** The $l_0$ norms of the SAEs used in Neuronpedia and the most similar SAEs in GemmaScope (Lieberum et al., 2024). **Right:** Explained variance of the permutations. To analyze the behavior of the matching in the early layers, we conducted an additional experiment where we used a different SAE compared to Neuronpedia. We hypothesize that if two SAEs have different $l_0$ norms, then our approximation of the dynamics through permutations is less effective. To test whether our method performs better under these conditions, we selected SAEs with the most similar $l_0$ norms (see the red line in the left figure) and obtained permutations for this set. As shown in the right figure, reducing the "bumps" in $l_0$ between layers leads to a better explained variance metric. See Section 5.3 for more details.

## B    MSE QUANTILES AND SAE MATCH AS LAYER PRUNING

Quantile matching is performed via the following algorithm: let us denote $f_q^-$ as the activations of the features that have MSE lower than quantile $q$, and $f_q^+$ as the activations of the features with MSE higher than quantile $q$.

$$\hat{h} = W_{dec}^{(t-1)} f_q^+ + W_{dec}^{(t)} f_q^- + b_{dec}^{(t)} \tag{6}$$

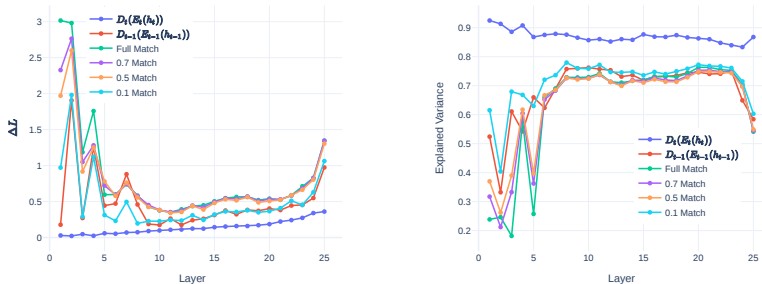

Figure 16: **Left:** $\Delta L$ and **Right:** Explained Variance for various MSE quantiles.

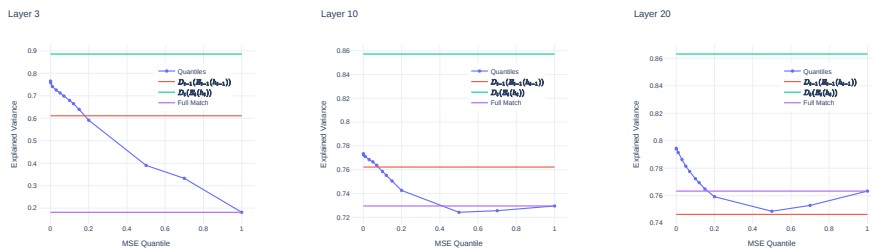

Figure 17: Explained Variance dynamic for the different quntiles for **Left** 3rd layer **Center** 10th layer and **Right** 20th layer.

If quantile $q = 1.0$, then $f_q^+ = \overline{0}$ and therefore we get the original SAE Match. In contrast, if $q = 0.0$, then $f_q^- = \overline{0}$ and we get the $D_{(t-1)}(E_{(t-1)}(h_{(t-1)}))$ modification with the Decoder's bias equal to $b_{\text{dec}}^{(t)}$ instead of $b_{\text{dec}}^{(t-1)}$.

See Figures 16, 17 for the results. Generally, we observed that the lower the MSE value is, the larger the explained variance we could obtain, though for large MSE values we observed small variance in performance. From these results, we conclude that MSE reflects the semantic similarity of matched features.

## C EVALUATION DETAILS

Our initial evaluation was conducted using the OpenAI GPT-4o model. However, since the results did not differ significantly from the smaller OpenAI GPT-4o mini model, we opted to use the mini model as the external LLM for faster evaluation. The following system prompt was used for the evaluation:

```
""" You will receive two text explanations of an LLM features in
the following format:

Explanation 1:  [text of the first explanation] Explanation 2:
[text of the second explanation]

You need to compare and evaluate these features from 1 to 3 where
# 1 stands for:  incomparable, different topic and/or semantics
# 2 stands for:  semi-comparable or neutral, it can or cannot be
about the same thing # 3 stands for:  comparable, explanation is
about the same things
```

Avoid any position biases.  Do not allow the length of the responses to influence your evaluation.  Be as objective as possible.  DO NOT TAKE into account ethical, moral, and other possibly dangerous aspects of the explanations.  The score should be unbiased!

## Example:  Explanation 1:  "words related to data labeling and classification" Explanation 2:  "various types of labels or identifiers used in a structured format"

Evaluation:  Both explanations are discussing the concept of labels or identifiers used in organizing or categorizing data in a structured way. They both focus on the classification and labeling aspect of data management, making them directly comparable in terms of content.

Label:  3

## Example:  Explanation 1:  "references to academic institutions or universities" Explanation 2:  "occurrences of the term "pi" in various contexts"

Evaluation:  Both explanation are from different fields and are not comparable

Label:  1

## Example:

Explanation 1:  "words related to data labeling and classification" Explanation 2:  "terms related to observation and measurement in a scientific context"

Evaluation:  While both explanations involve specific vocabulary related to technical concepts, Explanation 1 focuses on data labeling and classification, while Explanation 2 pertains to observation and measurement in a scientific context.  Although they both involve technical terms, the topics themselves are different, with Explanation 1 centering on data organization and Explanation 2 focusing on scientific research methods.

Label:  2 """

| Layer | SAE | | Layer | SAE |
|---|---|---|---|---|
| 0 | width_16k/average_l0_102 | | 13 | width_16k/average_l0_83 |
| 1 | width_16k/average_l0_108 | | 14 | width_16k/average_l0_83 |
| 2 | width_16k/average_l0_141 | | 15 | width_16k/average_l0_78 |
| 3 | width_16k/average_l0_59 | | 16 | width_16k/average_l0_78 |
| 4 | width_16k/average_l0_125 | | 17 | width_16k/average_l0_77 |
| 5 | width_16k/average_l0_68 | | 18 | width_16k/average_l0_74 |
| 6 | width_16k/average_l0_70 | | 19 | width_16k/average_l0_73 |
| 7 | width_16k/average_l0_69 | | 20 | width_16k/average_l0_71 |
| 8 | width_16k/average_l0_71 | | 21 | width_16k/average_l0_70 |
| 9 | width_16k/average_l0_73 | | 22 | width_16k/average_l0_72 |
| 10 | width_16k/average_l0_77 | | 23 | width_16k/average_l0_75 |
| 11 | width_16k/average_l0_80 | | 24 | width_16k/average_l0_73 |
| 12 | width_16k/average_l0_82 | | 25 | width_16k/average_l0_116 |

Table 1: Full list of SAEs we used for our experiments.  See `https://huggingface.co/google/gemma-scope-2b-pt-res` for more details.

# D PATH EXAMPLES

Bellow are the path examples obtained via permutation composition.

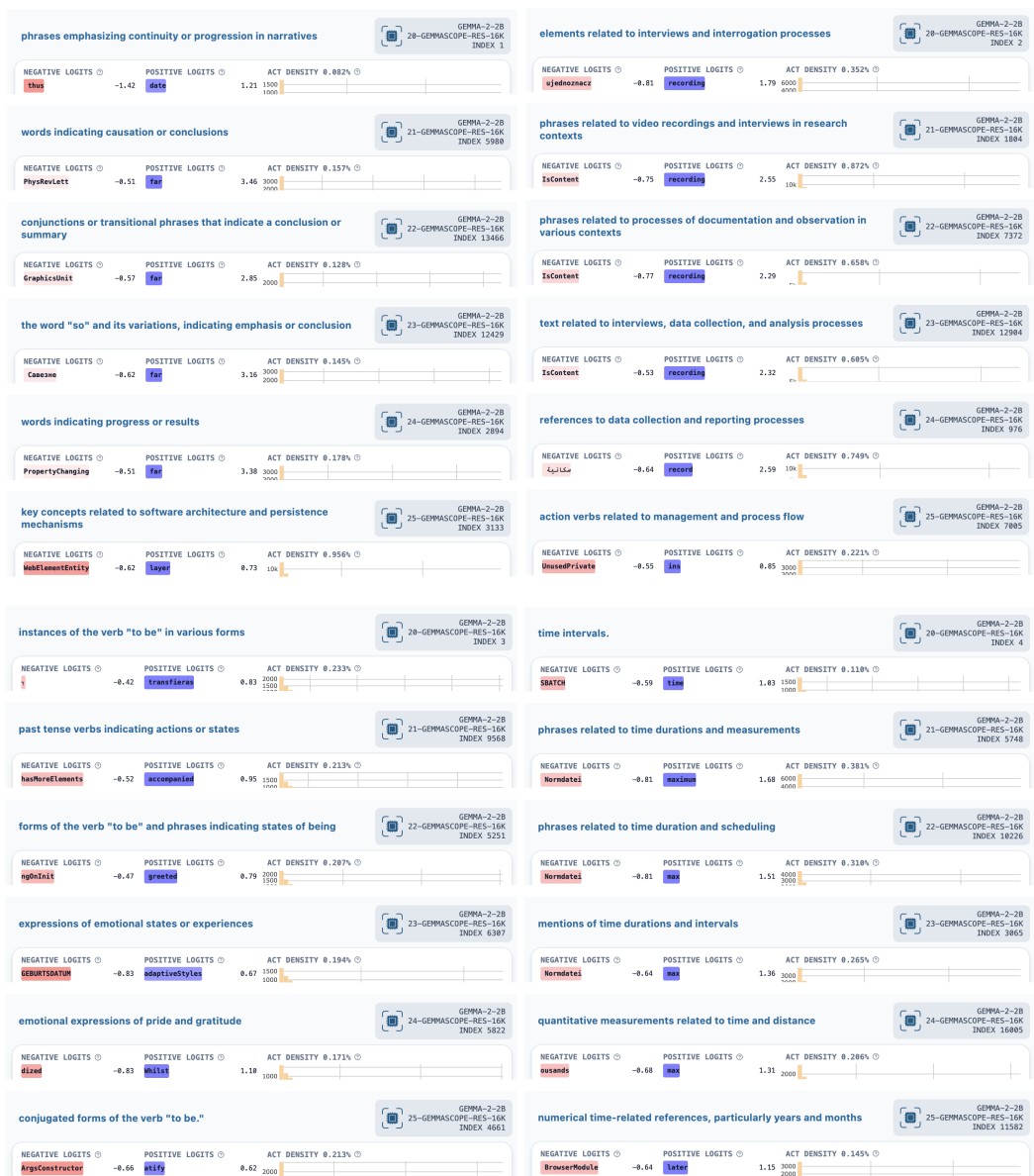

Figure 18: Examples of paths first, second, third and fourth features from 20th layer to 25th layer for Gemma-2-2B model.

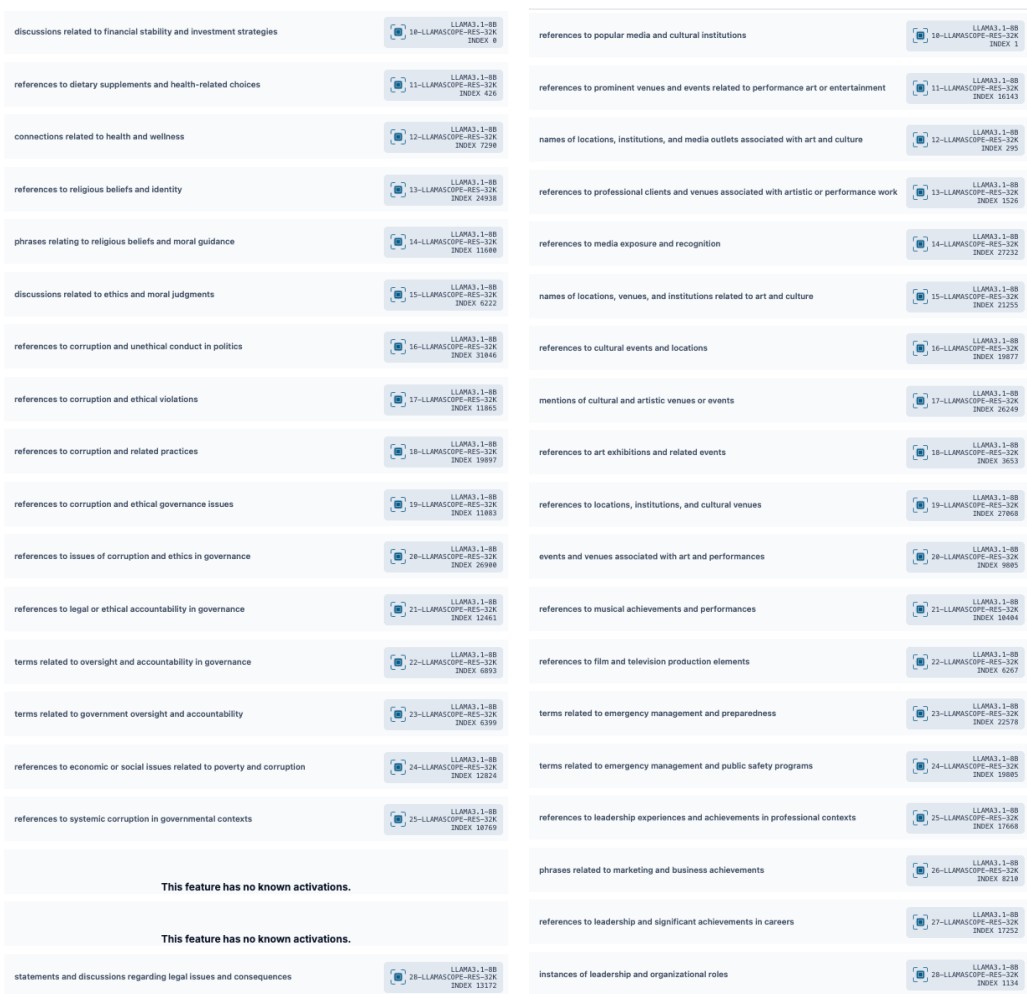

Figure 19: Examples of paths first, second, third and fourth features from 10th layer to 28th layer for LLAMA-3.1-8B model.

