# OpenReview forum: "Mechanistic Permutability: Match Features Across Layers"
_ICLR.cc/2025/Conference — ICLR 2025 Poster_

### Official Review · Reviewer_UKCQ · 2024-11-02

**Soundness:** 3
**Presentation:** 3
**Contribution:** 4
**Rating:** 8
**Confidence:** 4

**Summary:**

The paper presents a technique to match SAE latents from two SAEs across different layers called SAE Match. The technique finds permutations of latents which minimize MSE between the encoder and/or decoder representation of the feature in both SAEs. SAE Match addresses different layer norms by folding in the jumprelu threshold parameter into encoder and decoder matrices. The paper evaluates the technique on the Gemma Scope SAEs for Gemma 2.

**Strengths:**

Using MSE loss between SAE decoders is simple and fast. The idea of folding the jumprelu threshold parameter into weights to account for differences in layer norm is a really nice idea as well. Aligning SAE latents across layers is important for understanding how features progress throughout the foward pass of the model, so this work is useful to the field.

**Weaknesses:**

The paper only focused on a single model (Gemma) and a single type of SAE (JumpReLU). The paper evaluates on Gemma Scope, but uses only a single SAE per layer despite Gemma Scope containing multiple SAEs per layer with different sparsities. It would be a good check to see how the method performs comparing different SAEs trained on the same layer as well. The method seems like it does not address feature splitting, where a single latent in one SAE becomes multiple latents in a different SAE - the method seems like it will only pair latents 1-to-1.

**Questions:**

- In Figure 3, there are different results for folder and unfolded b_enc, but b_enc is not affected by folding according to equation 4. How is b_enc different due to folding?
- In Figure 3, vanilla matching performs not much worse than the actual matching techniques. Is vanilla matching essentially equivalent to randomly pairing decoder rows? If so, why is the MSE loss so low?
- In Section 4, it says SAEs with regularization coefficient near 70 are used. Is this referring to the L0 of the SAE in Gemma Scope? Neuronpedia only supports Gemma Scope SAEs with L0 closest to 100 - if Neuronpedia is used for all SAEs, does this mean that the L0 near 100 Gemma Scope SAEs are used as well?

---

> ### Author Response · Authors · 2024-11-21
>
> Thank you for your valuable review and for highlighting the importance of understanding feature dynamics.
>
> **Q:** In Section 4, it says SAEs with regularization coefficient near 70 are used. Is this referring to the L0 of the SAE in Gemma Scope? Neuronpedia only supports Gemma Scope SAEs with L0 closest to 100 - if Neuronpedia is used for all SAEs, does this mean that the L0 near 100 Gemma Scope SAEs are used as well?
>
> **A:** Thank you for bringing this to our attention. We have carefully validated that the features we observe match the descriptions in Neuronpedia. It appears that Gemma Scope updated the list of SAEs on October 22nd (see [commit link](https://huggingface.co/google/gemma-scope-2b-pt-res/commit/389d928eac6488b7052718d8ea03bf00357bcc63)). We have updated our paper in the rebuttal revision to align with these changes. Please refer to the red text in Section 4 for details.
>
> **Q:** In Figure 3, there are different results for folder and unfolded b_enc, but b_enc is not affected by folding according to equation 4. How is b_enc different due to folding?
>
> **A:** According to Eq. 4.  $b_{enc} = b_{enc} \odot \frac{1}{\theta}$ .
>
> **Q:** The paper only focused on a single model (Gemma) and a single type of SAE (JumpReLU). The paper evaluates on Gemma Scope, but uses only a single SAE per layer despite Gemma Scope containing multiple SAEs per layer with different sparsities. It would be a good check to see how the method performs comparing different SAEs trained on the same layer as well.
>
> **A:** Thank you for this suggestion. We have evaluated the proposed matching method by minimizing the MSE on the recently released Llama Scope. Please see Figures 13 and 20 in the rebuttal revision for the results.
>
> **Q:** The method seems like it does not address feature splitting, where a single latent in one SAE becomes multiple latents in a different SAE - the method seems like it will only pair latents 1-to-1.
>
> **A:** You are correct that our method establishes a bijective relationship between the latents of the SAE. We chose to approach the problem step-by-step, and therefore a bijective mapping is a natural starting point. Without first conducting this foundational work, it would be premature to experiment with more complex strategies for understanding feature dynamics, as we would lack a clear baseline. Our findings suggest that we should consider more general approaches (e.g., surjective or injective mappings) in future work.
>
> **Q:** In Figure 3, vanilla matching performs not much worse than the actual matching techniques. Is vanilla matching essentially equivalent to randomly pairing decoder rows? If so, why is the MSE loss so low?
>
> **A:** The MSE value of vanilla matching (and other strategies) depends on the norms of the weight columns (and rows) themselves. If the mean norm is sufficiently small, then the MSE of vanilla matching will also be small, although the MSE of the matching strategies will be lower. Please refer to Figure 16 in the rebuttal revision for norms at specific layers.
>
> We are open to further discussions and kindly ask that you consider reevaluating your score if our response has addressed your concerns.

---

> > ### Comment · Reviewer_UKCQ · 2024-11-24
> >
> > Thank you for the response. Given the changes to the paper and evaluating on another model, I've updated my score to 8.

---

### Official Review · Reviewer_hmcz · 2024-11-03

**Soundness:** 2
**Presentation:** 3
**Contribution:** 4
**Rating:** 5
**Confidence:** 3

**Summary:**

This paper proposes a new method for comparing and matching features of SAEs of nearby layers in transformers. They propose scaling features according to the activation threshold $\theta$ of a JumpReLU activation function in order to match features in a more natural norm for the underlying activations.

**Strengths:**

- Proposes a novel and interesting strategy for pairing features between layers.
    - Studies some of the shortcomings (e.g. long tail of pairing 'failures') of this strategy.
- The presentation is very clear and understandable.

**Weaknesses:**

- It would be good to spend more time justifying the hypothesies of Section 3. I do not think that the results in Figure 3 constitute much evidence for Hypothesis 2, since the reasoning here seems slightly circular - you propose parameter folding based off the observation that $\theta$ tracks the activation norms, but then evaluate feature similarity using the same objective that you are explicitly trying to minimize. Therefore, it is trivially true that 'folding+matching' outperforms 'matching'.
    - The evidence for Hypothesis 2 could be strengthened by including an analysis of how the scale of $\theta_i$ tracks with the scale of feature $i$ *on the same layer* (and similarly measuring MSE by scaling the features according to data-dependent statistics instead of $\theta$, e.g. by comparing MSE in the norm induced by some whitening transformation?); the observation in Figure 1 shows that *mean* $\theta$ tracks with *mean* activation norm, but your hypothesis rests on the assumption that $\theta_i$ is predictive of the scale of feature $i$ for features of the same layer.
    - It would be more convincing if you backed up your claims in Figure 2 that "reconstructions of matched features are closer to each other than in the unfolded variant of the algorithm" by comparing the reconstruction loss (perhaps again under a norm induced by a whitening transformation?) of 'folded+matched' and 'matched' permutations when 'skipping' a layer, as in Section 5.5.
- There was little analysis of the results of the experiments in Section 5.5. The loss differences resulting from using feature-matched activation reconstructions were not compared to any nontrivial baseline (e.g. perhaps something like a linear approximation of the layer?) and it was hard to see what the desired conclusion/hypothesis or suggestion at a direction for future study was here, and the paper might benifit from expanding on this a bit.
    - It might also be useful to baseline the figures in Figure 10 using the reconstructions of an SAE at layer $t+1$, i.e. substituting $x^{(t+1)}$ with $W_\mathrm{dec}^{(t+1)} \sigma ( W_\mathrm{enc}^{(t+1)} x^{(t+1)} + b_\mathrm{enc}^{(t+1)} ) + b_\mathrm{dec}^{(t+1)}$; it is known that SAEs achieve nonzero reconstruction loss and that this incurs a performance penalty when using the reconstructions in place of the original activations. Intuitively the penalty incured from using the matched activations is 'mixed' with this base reconstruction penalty.
    - As stated before, it would also be good to include the 'matched' baseline in this section as well.

**Questions:**

Could you provide more direct evidence for Hypothesis 2, or provide a clearer argument for why your current result provide evidence for it? Could you provide more discussion of what the takeaways for Section 5.5 were intended to be?

---

> ### Author Response · Authors · 2024-11-21
>
> Thank you for your valuable review and for highlighting the presentation and interesting aspects of our strategy.
>
> **Q:** It would be good to spend more time justifying the hypothesies of Section 3. I do not think that the results in Figure 3 constitute much evidence for Hypothesis 2, since the reasoning here seems slightly circular - you propose parameter folding based off the observation that
>
> **A:** Thank you for your feedback. We have improved the writing of Hypothesis 2 in the rebuttal revision to avoid circular justification. Indeed, Figure 3 does not justify Hypothesis 2 but rather demonstrates that the MSE value differs with and without folding. Instead, we now define the second hypothesis as "folding improves matching," which is clearly observable in Figure 4. Thank you for bringing this to our attention.
>
> **Q:** The evidence for Hypothesis 2 could be strengthened by including an analysis of how the scale of  tracks with the scale of feature  *on the same layer…*
>
> **A:** Since the decoder features are originally unit-normalized, the absolute value oftheta can be considered a proxy for the feature norm. You can find the mean absolute value of theta in Figure 2. Additionally, in Figure 16, we provide the norms of the decoder features after folding theta into the parameters.
>
> **Q:** It would be more convincing if you backed up your claims in Figure 2 that "reconstructions of matched features are closer to each other than in the unfolded variant of the algorithm" by comparing the reconstruction loss (perhaps again under a norm induced by a whitening transformation?) of 'folded+matched' and 'matched' permutations when 'skipping' a layer, as in Section 5.5.
>
> **A:** Thank you for the suggestion. We have added the experiment you requested. The results are presented in Figure 11.
>
> **Q:** There was little analysis of the results of the experiments in Section 5.5….
>
> **A:** Some of the suggested experiments are already included in the Appendix (see Figure 14). However, we noticed reviewers' interest in this experiment, so we have provided additional results in the rebuttal revision. We have marked the new results with red text for easy reference; please refer to them.
>
> **Q:** Could you provide more discussion of what the takeaways for Section 5.5 were intended to be?
>
> **A:** We view pruning as a natural way to evaluate matching strategies. This does not imply that matching could serve as a pruning method—a specifically designed pruning method would undoubtedly outperform pruning via matching. However, if we understand the dynamics of feature changes, then with sufficient evidence, we could skip some layers by replacing their evaluations with matched features.
>
> **Q:** It might also be useful to baseline the figures in Figure 10 using the reconstructions of an SAE at layer , i.e. substituting  with ; it is known that SAEs achieve nonzero reconstruction loss and that this incurs a performance penalty when using the reconstructions in place of the original activations. Intuitively the penalty incured from using the matched activations is 'mixed' with this base reconstruction penalty.
>
> **A:** You can find the reconstruction error in Figure 11, which shows (D_t(E_t(h_t))).
>
> We are open to further discussion if you have additional questions. If we have addressed your concerns, we kindly ask you to reevaluate your score.

---

> ### Author Response · Authors · 2024-11-25
>
> Dear reviewer hmcz,
> We would like to respectfully remind you that the discussion phase will be ending soon. We kindly ask you to please review the updated version of our paper and our responses to your questions. We are confident that we have addressed all of the major concerns you raised. If you agree, we kindly request that you reconsider the score you have given us. If there are still areas that need improvement, please let us know how we can enhance our work further.

---

> > ### Comment · Reviewer_hmcz · 2024-11-25
> >
> > Thank you for the updates and comprehensive reply, and apologies for the lateness of my reply.
> >
> > 1. It would be good to update or remove the Figure 3 caption and discussion in Section 5.1 to reflect this. In the caption, it still says that this experiment supports Hypothesis 2, which is untrue, and the corresponding discussion is still about a trivial result. It is unclear what Figure 3 is showing given that it does not support Hypothesis 2.
> >
> > 2. I think the claim "since the decoder features are originally unit-normalized, the absolute value oftheta can be considered a proxy for the feature norm" is non-trivial and should require empirical evidence. Figure 16 again only shows the average norm across depth, which is not relevant for my point.
> >
> > 3. Thank you for including this. This is good evidence for the usefulness of your algorithm over plain matching. It would still be useful to compare to cosine-similarity matching as a baseline.
> >
> > 4. It would be nice to have more discussion/description of Figure 13, as it took me a little while to figure out what it was showing, and the results do not seem positive, but this is not critical.
> >
> > 5. Thank you, this makes sense.
> >
> > 6. Again, thank you for including Figure 11.
> >
> > Overall, the inclusion of Figure 11 is good, but there remain some framing issues around Hypothesis 2 and the claim that "since the decoder features are originally unit-normalized, the absolute value oftheta can be considered a proxy for the feature norm" which I believe requires more evidence. I have raised my 'contribution' score accordingly, as I think Figure 11 is a useful result, but don't think that this warrants a change in my overall rating of the paper. If the disputed claim had more empirical evidence, I would raise my score to a 6. Thank you for your in-depth response!

---

> > > ### Author Response · Authors · 2024-11-25
> > >
> > > Thank you for your reply.
> > >
> > > > I think the claim "since the decoder features are originally unit-normalized, the absolute value oftheta can be considered a proxy for the feature norm" is non-trivial and should require empirical evidence.
> > >
> > > First, let us provide a citation from GemmaScope [1] (Section 3.2, top right of the 4th page):
> > >
> > > > Throughout training, we restrict the columns of W_dec to have unit norm by renormalizing after every update. We also project out the part of the gradients parallel to these columns before computing the Adam update, as described in Brickenet al. (2023).
> > >
> > > From this fragment, we can see that normalization was done for training purposes.
> > >
> > > We have now updated Figures 16 and 17 (colored purple for your convenience). As you can see, the norms of the $W_{dec}$ vectors across layers remain constant at 1 before folding. After folding is applied, norms from $\theta$ are transferred to $W_{dec}$ and $W_{enc}$ as formulated in Equation (4). We refer to vectors from $W_{dec}$ as **feature vectors** or simply **feature**.
> > >
> > > We hope that this answers your questions and clarifies any misunderstandings from our discussion. If you have any more questions, please feel free to ask in your reply.
> > >
> > > [1] [Gemma Scope: Open Sparse Autoencoders Everywhere All At Once on Gemma 2](https://arxiv.org/pdf/2408.05147)

---

> > > > ### Comment · Reviewer_hmcz · 2024-11-25
> > > >
> > > > I understand (and agree with) this line of argument. I have no problems with what you have written in your reply.
> > > >
> > > > You then go on to claim that this norm relates to the feature activation magnitude, which you show to be true in average (i.e. average $\theta$ of features at a layer corresponds to average norm of activations at that layer). The non-trivial part I am concerned about is how tightly the $\theta$ *of a given feature* corresponds to the average (which average you take is unclear to me here) activation *of that feature*, which you then use to justify your method. It could certainly be the case that $\theta$ is a good indicator for the average activation magnitude! However, it is not clear to me that this is immediately the case.

---

> > > > > ### Author Response · Authors · 2024-11-26
> > > > >
> > > > > Thank you for your reply. We have added the activation density histogram before and after folding, as seen in Figure 18. Folding makes these distributions more similar. Please let us know if you need any additional results, we will be happy to provide them.

---

> > > > > > ### Comment · Reviewer_hmcz · 2024-12-02
> > > > > >
> > > > > > Thanks for including this figure. It would be better to include a more standard scatter plot of (average?) activation norm to $\theta$-value (as it is not clear to me how much the distributions have been made similar/lower-variance, nor is this evidence of a consistent effect, as only 3 features are shown, and your claim is about a consistent effect), or to calculate normalized statistics of the activations before and after folding. Also, in Section 5.4, there seems to be an incorrect reference to Figure 18 now (it should point to some other Figure).
> > > > > >
> > > > > > Overall, I still feel that more evidence (like the figure I described above) is needed to strengthen Section 3.1.

---

### Official Review · Reviewer_mxq3 · 2024-11-04

**Soundness:** 3
**Presentation:** 4
**Contribution:** 2
**Rating:** 5
**Confidence:** 3

**Summary:**

The authors define a similarity metric between two learned encodings, based on permutations. They find that, specifically for the JumpReLU architecture, θ tracks the growing residual stream norm. Accounting for this observation by normalizing weight vectors with θ improves permutation matching. They find that W_dec MSE decreases substantially for high layers, while not making differences for early layers. They further employ an LLM to judge the semantic similarity of matched features. Interestingly, they find that feature matching works well in early layers (<10) as per MSE but show that feature descriptions do not match. Additionally, they compare matching features across multiple layers by exact match and layer-wise composition, and investigate SAE match as layer pruning technique.

**Strengths:**

Main finding: Cosine similarity alone is not a great proxy for late layers, as residual stream norms increase. The authors propose parameter folding, which effectively addresses this problem for JumpReLU SAEs. Current work relies on cosine similarity, and I am convinced the field should adopt this proposed technique.

**Weaknesses:**

### Critiques that can be addressed in this paper
- I am unsure whether the original hypothesis of permutation is answered. The term "matching" implies a binary measure of whether a feature mapping is true or false. This might require the introduction of a cutoff threshold, or applying a clustering technique. Otherwise, the framing of similarity measures might be clearer than permutation. I'm curious about the authors' opinion on whether there is a binary criterion for whether features do/don't match.
- To quantify the fraction of "true matches," the MSE over all features is too coarse; only the matching score distribution provides a clear picture of which fraction of features are "well enough" matches.
- Line 315: "...unfolded matching showed higher MSE in the scale of hidden state representations, supporting behaviour described with Hypothesis 2." Comment: This is not true for benc MSE.
- I'm unsure about the statistical significance of the LLM evaluation. I understand that 100 out of 16k (<0.1%) SAE latents (aka features) were chosen. Did the authors choose these features at random? Increasing the number of SAE features would increase the significance of their findings. Using an LLM to judge the coherence of max activating examples of features would be an insightful additional metric that would track the suspected polysemanticity in early layers.
- I do not agree that Figure 7 right reflects the findings in Figure 7 left, where the matching score suddenly increases at layer 10.
- The quantification of "too far apart" in Hypothesis 3 would be a useful improvement.

### Further suggestions (that can be addressed in future work)
How does the mechanistic permutability transfer to other SAE architectures? The only difference will be adapting scaling factors defined in Equation 4. It would be great to provide a generalized definition of parameter folding for all SAE architectures.

**Questions:**

### Questions
- I'm curious about the takeaways of the experimental results from Section 5. How do I interpret a scale of ∆L about 1? Do the authors believe their results indicate that layers between 10 and 20 can be collectively pruned with the SAE matching method?
- I'd be interested in a discussion of why encoder matching performs worse than decoder matching.

### Further notes
- Line 154: Repetition of meaning in two following sentences.
- Why is Hypothesis 2 formatted as a hypothesis? It seems like a definition of a method to me.
- Line 234: Calling "average ℓ0" a regularization coefficient is misleading; calling it (average) sparsity is clearer.
- Line 235: Do the authors refer to Equation 3 when mentioning MSE? A reference of that equation or a different naming would be useful, since MSE is often used for the reconstruction loss in the context of SAEs.
I- 'm very curious how these results compare to findings with crosscoders (https://transformer-circuits.pub/2024/crosscoders/index.html). Matched features might share decoder vectors of a single crosscoder feature.
- Section 5.3: Gurnee operates on MLP neurons, which are not incentivized to be sparse. I expect SAE features to be (more) monosemantic, so Gurnee's results ideally shouldn't apply to SAE features.
- I'd be curious about a deeper investigation of the sharp increase in semantic feature similarity in Figure 4 left.

---

> ### Author Response · Authors · 2024-11-21
>
> Thank you for your valuable review and for highlighting the importance of our proposed method.
>
> **Q:** I am unsure whether the original hypothesis of permutation is answered. The term "matching" implies a binary measure of whether a feature mapping is true or false. This might require the introduction of a cutoff threshold, or applying a clustering technique.
>
> **A:** Thank you for your question. We have added results incorporating thresholds to address this concern. Please refer to Appendix B in the rebuttal revision for details.
>
> **Q:** To quantify the fraction of "true matches," the MSE over all features is too coarse; only the matching score distribution provides a clear picture of which fraction of features are "well enough" matches.
>
> **A:** Thank you for pointing this out. Building on the results from the previous question, we believe that the MSE can serve as a threshold for matching, offering insights into the number of unmatched features.
>
> **Q:** Line 315: "...unfolded matching showed higher MSE in the scale of hidden state representations, supporting behaviour described with Hypothesis 2." Comment: This is not true for benc MSE.
>
> **A:** Thank you for your observation. In our latest revision, we have rewritten this section. Hypothesis 2 has been reformulated as "folding improves matching," which is clearly demonstrated in Figure 4. Additionally, the newly added Figure 11 shows that matching without folding results in lower explainable variance, further highlighting that folding improves quality.
>
> **Q:** I'm unsure about the statistical significance of the LLM evaluation.
>
> **A:** Thank you for raising this concern. Our LLM evaluation was not designed to provide statistically significant results but rather to offer a human-understandable assessment of a small subset of features. Nonetheless, the evaluations of delta loss and explainable variance exhibit consistent patterns, supporting the overall evaluation.
>
> **Q:** I do not agree that Figure 7 right reflects the findings in Figure 7 left, where the matching score suddenly increases at layer 10.
>
> **A:** Thank you for your feedback. The figures you mention actually evaluate different metrics—the left shows GPT-Score, and the right shows Matching Score. While they are not guaranteed to align perfectly, they do exhibit similar patterns.
>
> **Q:** The quantification of "too far apart" in Hypothesis 3 would be a useful improvement.
>
> **A:** Thank you for the suggestion. We have rewritten this section to provide a clearer quantification of "too far apart" in Hypothesis 3.
>
> **Q:** How does the mechanistic permutability transfer to other SAE architectures? The only difference will be adapting scaling factors defined in Equation 4. It would be great to provide a generalized definition of parameter folding for all SAE architectures.
>
> **A:** Thank you for bringing this up. We have evaluated the proposed matching method by minimizing the MSE on the recently released Llama Scope. Please see Figures 13 and 21 in the rebuttal revision for the results.
>
> **Q:** I'm curious about the takeaways of the experimental results from Section 5. How do I interpret a scale of ∆L about 1? Do the authors believe their results indicate that layers between 10 and 20 can be collectively pruned with the SAE matching method?
>
> **A:** Thank you for your question. We do not view SAE matching as a pruning method per se but consider pruning as a natural way to evaluate the quality of the matching compared to natural baselines. In our experiments, we prune one layer at a time to quantify the extent of dynamics we can approximate through permutations.
>
> **Q:** Section 5.3: Gurnee operates on MLP neurons, which are not incentivized to be sparse. I expect SAE features to be (more) monosemantic, so Gurnee's results ideally shouldn't apply to SAE features.
>
> **A:** Thank you for pointing this out. In our original version, we attempted to suggest that these results could be aligned. In the rebuttal revision, we have updated our hypothesis regarding the behavior of matching at the initial layers. Please refer to the red text in Section 5.3 for more details.
>
> We are open to further discussion if you have any remaining questions. If we have addressed your concerns, we kindly ask you to reevaluate your score.

---

> ### Author Response · Authors · 2024-11-25
>
> Dear reviewer mxq3,
> We would like to respectfully remind you that the discussion phase will be ending soon. We kindly ask you to please review the updated version of our paper and our responses to your questions. We are confident that we have addressed all of the major concerns you raised. If you agree, we kindly request that you reconsider the score you have given us. If there are still areas that need improvement, please let us know how we can enhance our work further.

---

### Official Review · Reviewer_3W8E · 2024-11-04

**Soundness:** 4
**Presentation:** 3
**Contribution:** 3
**Rating:** 8
**Confidence:** 4

**Summary:**

The paper introduces SAE Match, a technique that produces a bijection from [sparse autoencoder (SAE) features in layer n] and [SAE features in layer n+1], with the objective of reducing the average squared distance (i.e., mean squared error (MSE)) between a feature and its target. They also introduce parameter folding, which seeks to augment SAE Match to work on features with different scales. The paper shows via experimentation on Gemma Scope SAEs  that features have more similar LLM-derived interpretations if they have lower MSE, and that SAE Match with parameter folding results in more similar interpretations.

**Strengths:**

- The paper's SAE Match technique appears to work well at finding corresponding SAE features between layers, which can contribute to the goal of mapping a "feature circuit" (as in Marks et al. (2024) https://arxiv.org/pdf/2403.19647). This technique is data-free, meaning only the SAE weights are required, and not any model or SAE activations.

- The paper also provides useful empirical evidence that SAEs find features between layers that are simultaneously 1) close in the space of parameters, and 2) interpreted to have similar meaning.

**Weaknesses:**

- The authors get far worse results on layers 0-9 of the model than on layers 10-25, indicating that the technique may not fully generalize. The authors claim that this is to be expected, saying "This phenomenon aligns with findings from previous research. Gurnee et al. (2023) also reported increased polysemanticity in the early layers of neural networks." This explanation is unsatisfactory because Gurnee et al. (2023) were working with LLM neurons, not SAE features. Additionally, Cunningham et al. (2023) found that earlier layers were more interpretable (see Figure 2 in https://arxiv.org/pdf/2309.08600).

- The proposed technique of parameter folding is only defined for SAEs using the JumpReLU activation function, and it is not clear how could be adapted to over activation functions like ReLU or TopK.

**Questions:**

Questions for the authors:

1. Parameter folding serves as a form of normalization for encoder/decoder weights (as is mentioned in lines 156-157). What happens if one instead matches feature with encoder/decoder weights normalized to be unit vectors? This would be equivalent to maximizing cosine similarity instead of minimizing MSE, as ||x-y||^2=||x||^2+||y||^2 -2 ||x||*||y||*cossim(x,y).

2. One might expect the set of features to change across layers as the model processes information. In that case, the "correct" form for a matching might not be a bijection. Could one instead define a non-bijective "matching" via P(f)=argmin_i ||f-g_i||_2 where f is a feature in layer n, and {g_i}_i=1^k is the set of features in layer n+1? How does this compare to Feature Matching?

3. In Section 5.4, Figure 8, the y-axis is labelled "GPT Score". What is that metric and how was it calculated? Previous metrics have been a "GPT Score" on a scale of Different/Maybe/Same, or Matching Score in a range of 0-1, but this does not appear to be either of those.

4. In Section 5.5, it seems Layer Pruning may introduce error in two ways: 1) feature activations in layer N do not result in perfect reconstruction of the residual stream even for layer N, and 2) features in layer N do not perfectly match features in layer N+1. Did any experiment disentangle those effects? For instance, what is the \Delta L from replacing x with \hat x? This would provide useful context for the quantities shown in Figure 10.

5. Many SAEs have "dead" features, which presumably contribute little to the matching process. Could SAE Match be modified to exclude "dead" features, and if so how?



If it is permitted for authors to make revisions before the final submission, there are several small changes that could improve the quality of the paper:

- Line 93: The loss function as written is incorrect. The L2 term is squared, and what is written as L0 should be L1. See e.g. Equation 4 in (Cunningham et al, 2023) (https://arxiv.org/pdf/2309.08600).

- Line 132-133 (equation 3): The L2 norm in the argmin needs to be squared to get the mean *squared* distance.

- Lines 147-149 (equation 4): It appears that b_dec should be b_enc.

References:

Wes Gurnee, Neel Nanda, Matthew Pauly, Katherine Harvey, Dmitrii Troitskii, and Dimitris Bertsimas. Finding neurons in a haystack: Case studies with sparse probing. Trans. Mach. Learn. Res., 2023, 2023. URL https://openreview.net/forum?id=JYs1R9IMJr.

Hoagy Cunningham, Logan Riggs Smith, Aidan Ewart, Robert Huben, and Lee Sharkey. Sparse autoencoders find highly interpretable features in language models. In The Twelfth International Conference on Learning Representations, 2024. URL https://openreview.net/forum? id=F76bwRSLeK.

Samuel Marks, Can Rager, Eric J. Michaud, Yonatan Belinkov, David Bau, Aaron Mueller. Sparse Feature Circuits: Discovering and Editing Interpretable Causal Graphs in Language Models. 2024. https://arxiv.org/abs/2403.19647

---

> ### Author Response · Authors · 2024-11-21
>
> Thank you for your valuable review and for highlighting the useful empirical results we obtained. Here are the questions we can answer without additional experiments. We will add more experimental results and discussion in a few days in a separate response.
>
> **Q:** The authors get far worse results on layers 0-9 of the model than on layers 10-25… This explanation is unsatisfactory because Gurnee et al. (2023) were working with LLM neurons, not SAE features. Additionally, Cunningham et al. (2023) found that earlier layers were more interpretable.
>
> **A:** Thank you for highlighting this. We attempted to softly hypothesize a connection between our results and those of Gurnee et al. by stating that our observations align. In rebuttal revision (see Section 5.3, highlighted in red), we have updated this section and addressed the issues in the early layers.
>
> **Q:** What happens if one instead matches feature with encoder/decoder weights normalized to be unit vectors?
>
> **A:** Thank you for your question. We would like to clarify that the weights released by Gemma Scope come with unit-normalized vectors in the decoder layer. Therefore, the baseline "Matched" can be considered equivalent to the method you propose. While we agree that matching by cosine similarity is intuitive, our experiments show that adjusting the activation thresholds and measuring MSE yields better results (see Figure 4). Additionally, in Figure 12, we show that the decoder's feature vectors (unnormalized after folding) perform approximately the same. In that figure, we refer to the encoder weights and biases as W_enc and b_enc when discussing the "Encoder.”
>
> **Q:** In that case, the "correct" form for a matching might not be a bijection. Could one instead define a non-bijective "matching" via P(f)=argmin_i ||f-g_i||_2 where f is a feature in layer n, and {g_i}_i=1^k is the set of features in layer n+1? How does this compare to Feature Matching?
>
> **A:** Thank you for your insightful suggestion. We agree that a bijection may not be appropriate for matching dynamic features across different layers. However, our observations indicate that most features persist between layers t and t+1, and that using a bijection serves as a reasonable approximation. We decided to explore this approximation to assess its effectiveness and how its quality varies across different layers. We plan to enhance our model of feature paths in future work by accounting for cases where features originate from MLP or Attention mechanisms, or when features split or merge across layers. We believe that this step-by-step approach will ultimately lead to more insightful results.
>
> **Q:** In Section 5.4, Figure 8, the y-axis is labelled "GPT Score". What is that metric and how was it calculated? Previous metrics have been a "GPT Score" on a scale of Different/Maybe/Same, or Matching Score in a range of 0-1, but this does not appear to be either of those.
>
> **A:** Thank you for pointing out the need for clarification. In Figure 8, we use a scoring scale where **Different = 0**, **Maybe = 1**, and **Same = 2**. While we acknowledge that this grading scale is imperfect—since "Maybe" is exactly half of "Same"—we believe it provides a straightforward way to present the results across different layers and matching methods. We will add a clarification to the figure's caption to make this clear.
>
> Regarding the Matching Score, it measures the proportion of times matched features appear together, calculated as the number of matched features that appear divided by the total number of features that appear (i.e., #appeared_matched_features/#appeared_features).
>
> **Q:** In Section 5.5, it seems Layer Pruning may introduce error in two ways: 1) feature activations in layer N do not result in perfect reconstruction of the residual stream even for layer N, and 2) features in layer N do not perfectly match features in layer N+1. Did any experiment disentangle those effects? For instance, what is the \Delta L from replacing x with \hat x? This would provide useful context for the quantities shown in Figure 10.
>
> **A:** Yes, we conducted experiments that address exactly these concerns. Please refer to Figure 14. In this figure, the x-axis represents different SAEs trained with varying weights for the regularization term. Additionally, we note that there is a third source of error: (3) even when features i and j are matched across layers, the scale of the feature may increase or decrease after passing through the layer (i.e., the layer amplifies or attenuates the feature magnitude). We also provide additional results in Figure 11 that further explore these effects.

---

> ### Author Response · Authors · 2024-11-21
>
> **Q:** Many SAEs have "dead" features, which presumably contribute little to the matching process. Could SAE Match be modified to exclude "dead" features, and if so how?
>
> **A:** Yes, it is possible to modify SAE Match to exclude dead features. This can be done by removing the corresponding columns and rows in the cost matrix (W1^T W2) —specifically, see the last term of Equation 3—and then applying the linear assignment problem solver algorithm. Note that this permutation would no longer be a bijection. We did not consider this modification initially because we found that the proportion of dead features in the SAEs we use is less than 1% (see Figure 13 in [1]). We also intentionally considered the simplest variant of feature dynamics in our work to focus on the core aspects of our method.
>
> **Q:** Line 93: The loss function as written is incorrect. The L2 term is squared, and what is written as L0 should be L1. See e.g. Equation 4 in (Cunningham et al, 2023) (https://arxiv.org/pdf/2309.08600).
>
> **A:** We will add clarification about the different regularization methods in the final revision, as you correctly noted that SAEs with ReLU activations are typically trained with an L1 regularization term.
>
> **Q:** The proposed technique of parameter folding is only defined for SAEs using the JumpReLU activation function, and it is not clear how could be adapted to over activation functions like ReLU or TopK.
>
> **A:** For ReLU and TopK activation functions, we believe that the parameters are already effectively folded, so parameter folding is not necessary. The different scales across features arise due to the parameter theta in JumpReLU, which is not present in ReLU or TopK. We evaluated the proposed matching method by minimizing the MSE on the recently released Llama Scope [2]. Please see the revised Figures 13 and 20 for the results.
>
> Please feel free to reach out if you have any further questions.
>
> References:
>
> [1] Jumping Ahead: Improving Reconstruction Fidelity with JumpReLU Sparse Autoencoders. Senthooran Rajamanoharan and Tom Lieberum and Nicolas Sonnerat and Arthur Conmy and Vikrant Varma and János Kramár and Neel Nanda
>
> [2] Llama Scope: Extracting Millions of Features from Llama-3.1-8B with Sparse Autoencoders. Zhengfu He and Wentao Shu and Xuyang Ge and Lingjie Chen and Junxuan Wang and Yunhua Zhou and Frances Liu and Qipeng Guo and Xuanjing Huang and Zuxuan Wu and Yu-Gang Jiang and Xipeng Qiu.

---

> > ### Comment · Reviewer_3W8E · 2024-11-26
> >
> > To the authors,
> >
> > Thank you for your responses, and for your improvements to your paper, especially the new Figure 11!
> >
> > However, some of small mistakes in the paper I identified seem to be unaddressed in the 11/26 revision. I hope you will have a chance to make these small edits to improve the paper. These include:
> > 1. In your rebuttal you said "In Figure 8, we use a scoring scale where Different = 0, Maybe = 1, and Same = 2. While we acknowledge that this grading scale is imperfect—since "Maybe" is exactly half of "Same"—we believe it provides a straightforward way to present the results across different layers and matching methods. We will add a clarification to the figure's caption to make this clear." The caption for Figure 8 appears unchanged. Additionally, the 0-2 scoring scale you described cannot explain those images because your graphs often go above y=2.
> > 2. In lines 93-94, the reconstruction loss is based on the *squared* L2 norm, but you have written the unsquared L2 norm.
> > 3. Similarly, in Equation 3 (lines 155-157), you need the *squared* L2 norm in the middle expression.
> >
> > I believe my original score of 8 remains correct.

---

> > > ### Author Response · Authors · 2024-11-27
> > >
> > > Thank you for pointing that out! We have uploaded the updated version.

---

### Author Response · Authors · 2024-11-21
**General Response**

We would like to highlight the changes made in the rebuttal revision of our paper:

- **Updated the list of used SAEs** due to changes in Gemma Scope. This update affected only the two initial layers, and all relevant results have been reevaluated accordingly.
- **Reformulated the explanation for performance on initial layers in Section 5.3** to attribute it to differences in L0 norms between matched SAEs, and we have provided supporting results.
- **Added experiments evaluating different thresholds of matching based on MSE values**, which further support Hypothesis 1.
- **Expanded Section 5.5 with new experiments** to enhance our findings.
- **Included evaluation with LLaMa**, demonstrating that the proposed method can be applied not only to the Gemma model.
- **Made other minor improvements in wording** as suggested by the reviewers.

All changes are highlighted in red for clarity.

---

### Meta-Review · Area_Chair_Vq6X · 2024-12-21

**Metareview:**

The paper introduces a novel method for mapping Sparse Autoencoder features across adjacent layers, which demonstrates significant improvements in feature correspondence and interpretability compared to traditional cosine similarity metrics. Empirical results showcase its effectiveness with reduced MSE and semantic similarity evaluations, while visualizations and thorough discussions enhance the method’s clarity and accessibility. Reviewers highlight the potential applicability of SAE Match across diverse tasks, particularly in advancing sparse representation and interpretability research. On the other hand, focus on a single model (Gemma) and activation type (JumpReLU) limits generalization. Some claims, such as those related to Hypothesis 2, would benefit from stronger empirical evidence and comparison with alternative matching methods. Nevertheless, SAE Match is recognized as a valuable and innovative contribution, with significant potential to inform future advancements in feature alignment and complex model interpretability.

**Additional Comments On Reviewer Discussion:**

Rebuttal was useful, especially including the Figure 11.

---

### Decision · Program_Chairs · 2025-01-22

Accept (Poster)